# Vitamin D Supplementation in the Assessment of Cardiovascular Risk Factors in Overweight and Obese Children

**DOI:** 10.3390/medsci10030049

**Published:** 2022-09-05

**Authors:** Luca Pecoraro, Fulvio Nisi, Angela Serafin, Franco Antoniazzi, Luca Dalle Carbonare, Giorgio Piacentini, Angelo Pietrobelli

**Affiliations:** 1Pediatric Unit, Department of Surgical Sciences, Dentistry, Gynecology and Pediatrics, University of Verona, P.le Stefani, 1, 37126 Verona, Italy; 2Department of Medicine, University of Verona, 37126 Verona, Italy; 3Department of Anesthesia and Intensive Care Units, IRCCS Humanitas Research Hospital, 20089 Milan, Italy; 4Pennington Biomedical Research Centre, Baton Rouge, LA 70808, USA

**Keywords:** vitamin D supplementation, 25(OH)D, cardiovascular risk factors, childhood obesity, total cholesterol, HDL-C, LDL-C, ALT

## Abstract

Background: Childhood obesity is associated with cardiovascular-disease (CVD) risk factors, an unfavorable lipid profile and reduced levels of 25(OH)D. The aim of our study is to evaluate whether vitamin D supplementation may play a role in the assessment of the CVD risk factors in overweight/obese children and adolescents. Methods: We performed a retrospective observational study involving children (9–15 years of age) with a known diagnosis of overweight or obesity (BMI > 25) and decreased levels of 25(OH)D (<25 ng/mL), who underwent oral vitamin D supplementation (100,000 UI, one vial/month) for six months. The anthropometric parameters, 25(OH)D, serum lipids and ALT levels were measured at the beginning (T0) and after 6 months (T1). Results: Of the 58 patients recruited, 45 had an increase in the serum 25(OH)D levels after supplementation. Vitamin D supplementation was associated with a decrease in the serum levels of the total cholesterol (*p* = 0.009), LDL-C (*p* = 0.005) and ALT (*p* = 0.005), and an increase in HDL-C (*p* = 0.03). These results were confirmed when the correction for the body mass index (BMI) was applied. Conclusions: The favorable effect of vitamin D supplementation on the total cholesterol, LDL-C, HDL-C and ALT could transform these values into modifiable risk factors starting in early childhood, with beneficial effects on long-term health.

## 1. Introduction

Childhood obesity is associated with many comorbidities [1] and reduced levels of 25(OH)D [2]. It is well known that obesity is related to an unfavorable lipid profile and several cardiovascular risk factors [3]. An inverse association has been demonstrated between serum 25(OH)D and some cardiovascular-disease (CVD) risk factors, such as hypertension, dyslipidemia, abnormal glucose homeostasis and atherosclerosis [4,5,6]. Although vitamin D supplementation is apparently unrelated to a decrease in the risk of cardiovascular events in healthy children, this study aims to explore the potential role of vitamin D in CVD [7,8]. When examining obese children, the body mass index (BMI) represents a confounding factor that cannot be ruled out [9]. Inconsistent results have been found when assessing the relationship between vitamin D supplementation and a decrease in CVD risk factors in childhood obesity [9], while weak evidence supports the beneficial effects of supplementation in insulin resistance in children with obesity [8]. Regardless of the conflicting evidence regarding vitamin D supplementation, there is a strong association between reduced serum 25(OH)D and childhood obesity [2]. Specifically, research has shown an inverse relationship between serum 25(OH)D and the BMI [2]. Moreover, serum 25(OH)D decreases by 1.15 ± 0.55 nmol/L per each 1% increase in the total fat mass [10]. However, it has not been demonstrated whether serum 25(OH)D insufficiency should be considered as a cause or consequence of obesity due to the fact that is not possible to discern cause and effect [11]. On the one hand, the decrease in serum 25(OH)D and the consequent increase in the parathyroid hormone would stimulate the production of vitamin D in an active form, with the enhancement of lipogenesis and a reduction in lipolysis, suggesting the role of this hormonal imbalance in the predisposition towards obesity. On the other hand, vitamin D is a lipophilic hormone, and it becomes unavailable in the blood stream to perform its biological functions because it is stored in adipose tissue [2,12]. Another hypothesis regards hepatic steatosis, which is also widespread in obese children and would reduce vitamin D hydroxylation by decreasing the serum 25(OH)D concentration [13]. Lastly, research has shown that poor lifestyle habits are present among subjects with obesity [11]. Serum 25(OH)D can also be reduced by dietary intake and a sedentary lifestyle, which, in turn, limits sun exposure and the skin synthesis of this vitamin [11]. The conflictual relationship between vitamin D and childhood obesity is shown by the inability to restore normal serum 25(OH)D levels in obese children through standard supplementation. It seems that higher doses are required in these individuals to compensate for the sequestration in adipose tissue [14]. Specifically, a daily intake of 600 IU is recommended for healthy children between 1 and 18 years of age, and doubling or tripling this daily dose is recommended for obese individuals or subjects taking medications such as anticonvulsants, antiretrovirals, antifungals or corticosteroids [15]. The aim of the present study was to investigate the relationship between vitamin D and obesity in pediatric subjects. In this study, the potential effect of vitamin D supplementation on the serum total cholesterol, HDL-C cholesterol, LDL-C cholesterol, triglycerides, ALT, APO A1 and APO B was investigated, keeping in mind the assessment of the cardiovascular risk factors in these patients.

## 2. Materials and Methods

A retrospective observational study was performed including children aged 9 to 15 years with a known diagnosis of overweight or obesity (BMI > 25) and decreased levels of serum 25(OH)D (<25 ng/mL), who attended the Women and Children’s Hospital in Verona, Italy, between 1 January 2015 and 31 December 2015. The starting point was represented by our clinical practice of dosing serum 25(OH)D in children affected by overweight/obesity at the moment of the diagnosis. If the diagnosis of hypovitaminosis D was confirmed, then patients underwent an oral vitamin D supplementation (100,000 UI, one vial/month) for six months, based on the evidence that overweight/obese children are at risk of poor compliance and an increased dose of vitamin D is recommended for these individuals [2]. In the context of this retrospective study, the following information was examined: (1) the patient’s personal data sheet (gender, date of birth and ethnicity) and anamnestic data sheet, regarding comorbidities known by the parents/guardians (one or both, if present); (2) a medical examination performed by a physician, including the investigation of the patient’s medical history, a medical evaluation, the measurement of the anthropometric parameters (height in centimeters, weight in kilograms) and the calculation of the BMI corrected for age; (3) venous blood sampling for laboratory analysis concerning the serum levels of 25(OH)D and ALT and the lipid profile (i.e., serum total cholesterol, LDL-C cholesterol, HDL-C cholesterol, triglycerides, APO A1, APO B), performed after fasting for at least 6 h. The medical examination, the measurement of the anthropometric parameters, the calculation of the BMI and the venous blood sampling for laboratory analysis were carried out upon diagnosis of overweight/obesity and hypovitaminosis D (T0) and after six months (T1). The serum 25(OH)D levels were analyzed using the LIAISON^®^ 25 OH Vitamin D Total Assay, and they could be considered “sufficient” (30–100 ng/mL), “insufficient” (21–29 ng/mL) or “deficient” (≤20 ng/mL) [16,17]. The total cholesterol, HDL-C and triglycerides were measured by an enzymatic technique, using Roche Cobas 6000^®^ (Roche Diagnostic, Basel, Switzerland). The quantitative determination of the human serum Apo B and Apo A1 was obtained by kinetic nephelometry with IMMAGE^®^ immunochemistry systems and APO CAL (calibrator for apolipoproteins). The ALT parameter was obtained with tests performed on the analytics platform COBAS 6000^®^ (Roche Diagnostics). 

This retrospective study was approved by the local Ethics Committee (code: GP-DVD-ITA13 (31CESC)). Patients were reported anonymously, and their parents signed written consent at the moment of the diagnosis of overweight/obesity and hypovitaminosis D. Data are reported as mean ± standard deviation or median and the IQR or number and percentage, as appropriate. The normality of the distribution was tested with a Kolmogorov–Smirnov test. A Student’s *t*-test was used for the parametric variables (i.e., serum total cholesterol, LDL-C cholesterol, HDL-C cholesterol, triglycerides, APO A1, APO B). Single and multiple linear regressions, together with the correlation coefficient (r) and determination coefficient (R2) were used to explore the association between the BMI, serum 25(OH)D, ALT and the lipid profile. The statistical significance level was set to *p* < 0.05. Analysis was performed with Microsoft Excel v.14.5.2 for Mac 2011, and R v.3.3.0 (“Supposedly Educational”) for Mac 2011.

## 3. Results

Fifty-eight individuals (35 M, 23 F; mean age: 12.11; SD ± 2.42) were deemed suitable for the study. The anthropometric and laboratory parameters of the patients are represented in Table 1. 

Out of the 58 enrolled patients, 45 showed an increase in serum 25(OH)D after supplementation. The statistically significant findings were the decrease in the serum total cholesterol (*p* = 0.009), as well as LDL-C cholesterol (*p* = 0.005) and ALT (*p* = 0.005), and an increase in the serum HDL-C cholesterol (*p* = 0.03). These data were further analyzed with correlation and determination coefficients. The correlation coefficients (r) and determination coefficients (R2) for the total cholesterol, LDL-C, HDL-C and ALT are shown in Table 2. 

The total cholesterol showed an inverse association with serum 25(OH)D in subjects with 25(OH)D insufficiency compared with those with a deficiency at the moment of the diagnosis of overweight/obesity and hypovitaminosis D (T0). When the supplementation was administered, the overall total cholesterol at (T1) was much lower than at (T0), and it was far from the threshold for cardiovascular risk (Figure 1A).

When the correlation total cholesterol–serum 25(OH)D was corrected for the BMI, the multivariate-linear-regression plot showed an increase in the total cholesterol related to a high BMI, and above all, a decrease in the BMI when the serum 25(OH)D levels increased (Figure 2A). This association was confirmed both at T0 and T1. There was an inverse linear correlation relating HDL-C cholesterol and serum 25(OH)D at T0, while a direct correlation was demonstrated at T1 (Figure 1B). High BMI values were associated with a decrease in the HDL-C levels when examining the multivariate-linear-regression plot corrected for the BMI. It is worth noting that the highest HDL-C values occurred when the BMI values were in the lower range and the serum 25(OH)D in the highest one (Figure 2B). As the serum 25(OH)D increased, there was a slight decrease in the LDL-C values at T0. An overall decrease in the LDL-C cholesterol values was found at T1. Specifically, there was a strong correlation (R^2^ = 0.89) between the increase in the serum 25(OH)D and the decrease in the LDL-C cholesterol values (Figure 1C). An inverse linear correlation regarding the ALT levels was observed at T0: specifically, as serum 25(OH)D increases, ALT tends to decrease (Figure 1D). During the assessment at T1, an overall decrease in the ALT values was observed. A slight decrease in the ALT serum levels was observed as the serum 25(OH)D levels increased after vitamin D supplementation. The comparison is significant if the trend of the reduction in the ALT values is compared before and after supplementation (Figure 1D). When the association is controlled for the BMI, the multivariate-linear-regression plot shows that the ALT values tend to decrease as the BMI increases both at T0 and T1. Serum 25(OH)D does not seem to produce considerable changes in the ALT levels (Figure 2C). 

Vitamin D supplementation was not associated with statistically significant findings when serum Apo A1 (*p* = 0.18), Apo B (*p* = 0.18) and triglycerides (*p* = 0.31) were examined.

## 4. Discussion

This retrospective study aimed to demonstrate the possible role of vitamin D supplementation in the assessment of the cardiovascular risk factors in overweight and obese children affected by low serum 25(OH)D. Vitamin D supplementation was associated with significant changes in the serum levels of the total cholesterol, LDL-C, HDL-C and ALT. These results were confirmed when the correction for the BMI was applied. Specifically, vitamin D supplementation was associated with a reduction in the total cholesterol, which is possibly driven by a significant reduction in the LDL-C and ALT levels. Moreover, it was associated with an increase in the HDL-C values, which were higher when the serum 25(OH)D was high, and the BMI was low. Another interesting finding is that the reduction in the BMI seemed to play a role, together with serum 25(OH)D, in reducing the values of the total cholesterol. Vitamin D supplementation was not associated with significant changes in serum Apo A1, Apo B and triglycerides; hence, no conclusions could be drawn. 

The current research regarding the relationship between serum 25(OH)D and the CVD risk factors in overweight and obese children is represented by conflicting evidence. Specifically, no significant changes in the total cholesterol have been reported in the literature with serum 25(OH)D variations [9,18,19,20,21,22]. Moreover, an association between changes in serum 25 (OH)D and serum LDL-C has not been demonstrated [18,19,22,23,24]. While Creo et al. [18] and Birken et al. [24] did not find an association between the serum levels of HDL-C and 25(OH)D, Iqbal et al. [22] and Williams et al. [23] found an association. The association between serum 25(OH)D and serum ALT is also not clear. Specifically, Creo et al. [18] and Naderpoor et al. [25] demonstrated no association, while Park et al. [26] did. The relationship between vitamin D supplementation and the cardiovascular risk factors in overweight and obese subjects is also conflicting. In general, there is no clear consensus regarding the role of vitamin D in childhood CVD [27]. Brezinski et al. [28] and Nassar et al. [29] found that vitamin D supplementation has no effect on the body-weight reduction in overweight and obese children with 25(OH)D insufficiency. Vinet et al. [30] found that vitamin D supplementation attenuated microvascular dysfunction in childhood obesity, based on evidence that this illness is related to endothelial dysfunction [31,32] To the best of our knowledge, there are no studies that investigate the association between vitamin D supplementation and the lipid profile in childhood obesity. Sacheck et al. reported that vitamin D supplementation had positive effects in normal-weight children on the HDL-C cholesterol, LDL-C cholesterol and total cholesterol, and especially at the dosage of 600 IU/d, with several significant changes persisting during the post-supplementation period [33]. On the contrary, Cai et al. established that vitamin D supplementation did not affect the HDL-C-C, LDL-C-C, total cholesterol, blood pressure, waist circumference and BMI in normal-weight children [34]. In general, the use of vitamin D supplementation for the improvement in cardiometabolic health is not indicated in childhood [8]. Weak evidence has suggested the beneficial effects on insulin resistance in children affected by obesity, but a possible unfavorable effect on serum LDL-C should be investigated in these patients [8].

This study demonstrated that vitamin D supplementation was associated with a reduction in the serum ALT levels. At the same time, the current research shows that elevated serum ALT levels in children are often indicative of abnormalities in the liver histology, and low serum 25(OH)D levels correlate with the severity and progression of nonalcoholic hepatic steatosis in children as well as adults. In addition, animal studies report how vitamin D supplementation may have a beneficial effect on both reducing the serum ALT and the reversibility of hepatic steatosis [35,36]. Looking at the reduction in the ALT levels by supplementation with vitamin D reported in this study, an improvement in liver function could be hypothesized, thus limiting the subsequent evolution towards hepatic fibrosis.

This study has some limitations. First, it is a retrospective study with major limitations in the quality and quantity of the data available for analysis because they were not collected according to the needs of the study. For many parameters, the absence of data assessment either at T0 or T1 restricted the sample number. Second, the type of diet, sun exposure and amount of physical activity performed in hours was not evaluated. These aspects could affect the variation in the serum 25(OH)D and BMI, and consequently, the cardiovascular risk factors. Finally, a specific body-composition assessment testing the fat mass and fat-free mass was not performed.

## 5. Conclusions

This study is the first to investigate the effect of vitamin D supplementation on the CVD risk factors in pediatric subjects affected by childhood obesity. The favorable effect on the total cholesterol, LDL-C and HDL-C could transform these lipidic values into modifiable risk factors starting in early childhood, with beneficial effects on long-term health. In addition, the reduction in the serum ALT levels by vitamin D supplementation could indicate an improvement in the liver function, thus limiting the subsequent evolution toward hepatic fibrosis. From the results of this study, performing examinations to assess the cardiovascular risk parameters and serum 25(OH)D could represent a starting point for vitamin D supplementation in overweight and obese children and adolescents affected by hypovitaminosis D. The results of this study could furthermore support the role of vitamin D supplements in improving the levels of the total cholesterol, HDL-C, LDL-C and ALT.

## Figures and Tables

**Figure 1 medsci-10-00049-f001:**
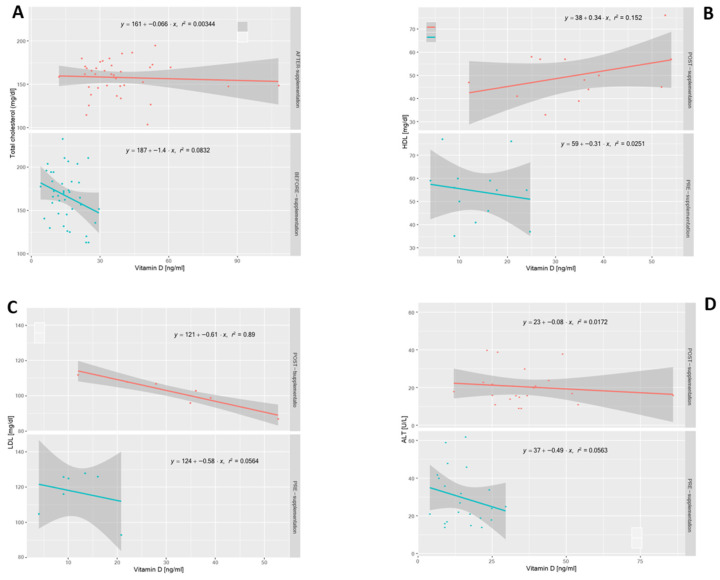
Linear-regression vitamin D-specific laboratory parameters before and after vitamin D supplementation: (**A**) total cholesterol; (**B**) HDL-C; (**C**) LDL-C; (**D**) ALT.

**Figure 2 medsci-10-00049-f002:**
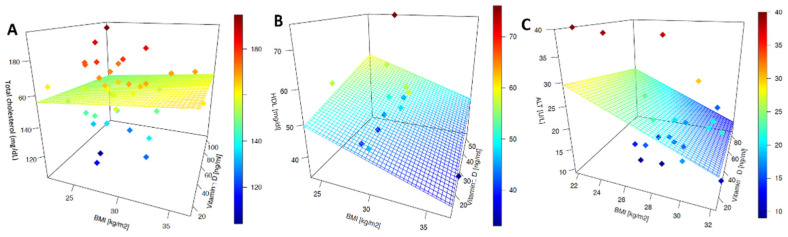
Vitamin D-specific laboratory-parameter multivariate linear regression (corrected for BMI) before and after vitamin D supplementation: (**A**) total cholesterol; (**B**) HDL-C; (**C**) ALT.

**Table 1 medsci-10-00049-t001:** Average and standard deviations (SDs) of anthropometric and laboratory parameters before and after vitamin D supplementation.

	T_0_		T_1_		*p*
	Average	±SD	Average	±SD	
WEIGHT (kg)	74.38	17.05	72.01	13.05	NA
HEIGHT (m)	154.52	9.99	158.79	8.39	NA
BMI (kg/m^2^)	30.78	4.45	28.35	3.18	NA
SERUM 25(OH)D (ng/mL)	16.22	6.52	32.65	16.68	NA
TOTAL CHOLESTEROL (mg/dL)	16.56	29.27	15.8	19.54	0.009
HDL (mg/dL)	54.32	12.40	50.15	10.57	0.03
LDL (mg/dL)	117.03	12.35	101.71	7.85	0.005
ALT (U/L)	29.64	14.15	20.27	8.93	0.005
APO B (g/L)	0.77	0.17	0.75	0.16	0.18
APO A1 (g/L)	1.41	0.15	1.38	0.17	0.18
TRIGLYCERIDES (mg/dL)	82.76	35.33	80.36	35.41	0.31

**Table 2 medsci-10-00049-t002:** Correlation coefficients (r, R2) between anthropometric and laboratory variables before and after vitamin D supplementation in patients with an increase in serum 25(OH)D after vitamin D supplementation.

	Coefficient r at T_0_ Correlation before Treatment	Coefficient r at T_1_ Correlation after Treatment	Coefficient R^2^ at T_0_	Coefficient R^2^ at T_1_
BMI–total cholesterol	0.07	0.10	0.006	0.009
BMI–vit D	0.05	0.06	0.003	0.003
Vit D–total cholesterol	−0.29	−0.06	0.08	0.003
BMI–HDL	−0.41	−0.38	0.17	0.150
BMI–vit D	−0.25	−0.30	0.06	0.090
Vit D–HDL	−0.16	0.39	0.03	0.150
BMI–LDL	0.50	0.46	0.26	0.210
BMI–vit D	−0.25	−0.29	0.06	0.090
Vit D–LDL	−0.24	−0.94	0.06	0.890
BMI–ALT	−0.26	−0.38	0.07	0.140
BMI–vit D	0.28	0.10	0.08	0.010
Vit D–ALT	−0.24	−0.13	0.06	0.020

## Data Availability

Not applicable.

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
