# Peer review of "Vitamin D Supplementation in the Assessment of Cardiovascular Risk Factors in Overweight and Obese Children"

_medsci, 2022, doi:10.3390/medsci10030049_

Round 1

Reviewer 1 Report

This is an interesting paper on the effects of vitamin D supplementation on CVD factors in children. However there are some major concerns regarding the methodology of the study.

There is a lot of information missing from the protocol description. First of all, the study is being described as a retrospective study but is not clear whether this is retrospective or a clinical trial. It seems that patients were recruited following specific inclusion and exclusion criteria and in some cases signed a consent but no bioethics committees’ approvals are mentioned. Nevertheless, if this is indeed a retrospective study the authors should at least report an approval form the IRB or some hospital authority for use of patient records or an institution consent for use of information of anonymous patients retrospectively. Also, if they have signed a consent for another reason (part of another study) this should be mentioned and the approval numbed should be stated. Also, Materials and Methods section is very short and data collection is not described in detail (more information needed on anthropometrics, biochemical measurements, blood collection procedures etc). Finally, the statistical approach is poor as regression analysis lacks important coefficients, such as sex and age or time of year (sun exposure is important for vitamin D levels) and sample size is considerably small, and no power calculation is mentioned.

Minor comments

1) The title would be more appropriate if it referred to effect on CVD factors, “prevention of early vascular complications” is not exactly the subject of this study

2) There is an affiliation missing (number 4)

3) You do not mention the reasons why vitamin D could have increased in only 45 out of 58 patients

4) When you refer to vitamin D in general you could use the term “vitamin D” instead of 25(OH)D. When referring to other studies you could be more specific on what type of vitamin D form they measured.

5) Please use the term plasma not plasmatic levels

6) Line 70. …”exploring” a potential role on prevention

7) Line 97. T-test to compare….?

8) line 97 single not sigle

9) In table 1 you should add the vitamin D levels

10) In table 2 you should add the p values

11) line 155. Prevention of CVD factors. Please rephrase

Author Response

uploaded as Word file

Reviewer 2 Report

The manuscript submitted by Pecorano et al., titled: "Vitamin D supplementation in the prevention of early vascular complications in overweight and obese children" is an interesting study investigating the relationship between vitamin D and vascular complications in obese children. 

The reviewer would like to offer the following points toward the improvement of the manuscript:

1. While the BMI is a measure/risk factor associated with health and disease it is important to note that in underage population the standard BMI scale cannot be used but rather the BMI for age calculation needs to take place and be used instead, especially when there is a range of age in the underage population (<18 yrs) and even more so when we have both girls and boys where the developmental stages, rates and hormonal profiles may differ significantly since they have not reached developmental maturity. Here is some information by CDC pertinent to the comments above: https://www.cdc.gov/healthyweight/bmi/calculator.html  

3. Furthermore, while BMI is an indicator of risk the most critical health/disease related element is actually body composition. Have the authors considered body composition? Especially if there are different levels of physical activity, different stages of growth and different sex in the participant group.

4. BMI is a unit-less measure. It is an index and is calculated by dividing the weight in kg over the heigh in m squared. This is not to be confused by generating a kg/m2 unit which would allude to pressure (force over surface) and has no physical meaning in the context of BMI. Moreover, we do not measure surface area as the unit m2 implies. So please refrain from using units for BMI.

5. Were all children of the same sex and what was that?

6. How was the number of participants selected? Was there a power calculation for example? Please justify the N in the study.

7. Please consider providing more information in terms of the demography and the selection criteria of your group.

8. What were the potential confounding factors. For example was diet considered? Diet can have a significant effect in blood biochemistry. Especially in terms of vitamin D status. While this is briefly mentioned as a limitation of the study the reviewer finds that this is a very critical major issue with the study because it creates an issue of normalization.  

Author Response

uploaded as Word file

Reviewer 3 Report

Major comments:

1. Please be more specific when you use the term "vitamin D", i.e. indicate, whether vitamin D3, 25(OH)D3 or 1,25(OH)2D3 is meant.

2. Please explain in the text why monthly supplementation was chosen, although in general daily supplementation is considered more efficient. Why a dose of 100,000 units, this would be more than 3000 Units/day and is per kg body mass a very high dose.

3. What was the problem with the 13 children, where even the very high bolus had no effect on 25(OH)D serum levels?

4. Fig. 1 is not very clear and should be improved on multiple levels, such as scale of graph, labeling and size of font. It may be far clearer, if the change of measured physiological parameter (the ratio rather than the delta) is plotted over the change of vitamin D status.

5. Fig. 2 is a totally different style as Fig.1. Both should be harmonized and same points as for Fig. 1 should be implemented.

Minor comments:

1. All abbreviations should be defined at first time use and then applied consistently. This applies also to the Abstract. Please do not define abbreviations twice.

Author Response

uploaded as Word file

Round 2

Reviewer 1 Report

All comments have been addressed succesfully

Author Response

We thank the REVIEWER 1. We have improved the English language with the help of a native English speaker.

Reviewer 2 Report

The authors have made a reasonable effort to address the reviewer's comments. Nonetheless the reviewer in his personal scientific opinion finds the diet/supplement information lacking a significant drawback and a non-addressed confounding factor.

Author Response

We thank the REVIEWER 2. We have noted these aspect into the section “limits of the study”. Moreover, we have improved the English language with the help of a native English speaker.

Reviewer 3 Report

The authors should take care that figures 1 and 2 are readable.

Author Response

We discussed with the statistician this suggestions and we think that the figures are suitable showing Vitamin D-specific laboratory parameters linear and multivariate linear regression (corrected for BMI) before and after vitamin D supplementation. Anyway, if the reviewer 3 and the editor deem it appropriate, we are eventually available to remove figures 1 and 2 from the article.